# Preventing Privacy Leakage in Vision-Language Models: A Secure Framework for Large-Scale Image Classification

## Abstract

Recently, large vision-language models (LVLMs) have demonstrated strong performance in generating pseudo-labels for diverse downstream tasks. However, during annotation or label generation, these models may inadvertently access sensitive information contained in the data (e.g., medical conditions, smoking habits), thereby creating potential risks of individual privacy leakage. To mitigate this challenge, we propose a novel framework that prevents LVLMs from accessing data associated with sensitive information. Specifically, our framework integrates a privacy label set with a randomized label set. Human annotators first determine whether the merged label set contains the ground-truth label; only when it does not, the LVLMs are employed to generate a pseudo-label. This mechanism ensures that LVLMs never directly access samples associated with sensitive information during annotation, while the inclusion of the randomized label set provides partial supervision for non-privacy samples. Moreover, we introduce a risk-consistent estimator that enables effective learning from LVLM-generated pseudo-labels under the exclusion of sensitive data. Extensive experiments on benchmark datasets demonstrate the superiority of our approach over state-of-the-art methods, effectively safeguarding sensitive label information while maintaining competitive model performance. Code is available at: https://anonymous.4open.science/r/VLMPrivacy-C468/

## 1 Introduction

Deep learning models have achieved remarkable advances in image classification, primarily driven by large-scale, manually annotated datasets (Robinson et al., 2024; Adcock et al., 2024; Cinquin et al., 2024; Joshi et al., 2025; Liu et al., 2025). However, acquiring such extensive annotations is often prohibitively expensive, and in many real-world scenarios, even infeasible (Wu et al., 2024; Xia et al., 2023; Li et al., 2025a; Demirel & Holz, 2025). To address this limitation, large vision-language models (LVLMs) have been increasingly employed to generate pseudo-labels as substitutes for manual annotations (Sun et al., 2022; Zhang et al., 2024; Xing et al., 2024; Hu et al., 2024). Trained on extensively cleaned web data and synthetic data, LVLMs exhibit strong generalization capabilities in image classification across diverse domains. Consequently, a widely adopted strategy to reduce manual annotation overhead involves uploading datasets to online LVLMs for pseudo-label generation (OpenAI, 2023; Reid et al., 2024; Botev et al., 2024; OpenAI, 2025).

However, this process inevitably exposes the entire dataset to the LVLMs, as illustrated in Figure 1 (a). Real-world visual data often contain highly sensitive information, such as personal identities, medical records, or other confidential content, posing substantial privacy risks (Wang et al., 2025; Guo et al., 2025). When such private information is uploaded to LVLMs, it is impossible to guarantee that online proprietary models will not misuse the data in ways that compromise user privacy, thereby raising serious ethical and security concerns (Chen et al., 2023; Mireshghallah et al., 2024; Guo et al., 2025). Thus, a critical question arises: How can we reduce the overhead of manual annotation while simultaneously preventing the leakage of sensitive information to LVLMs?

To answer this question, we propose a novel setting, called Privacy-Masked Labels (PMLs), to prevent the exposure of sensitive data to the LVLMs while reducing the manual annotation cost. Specif-

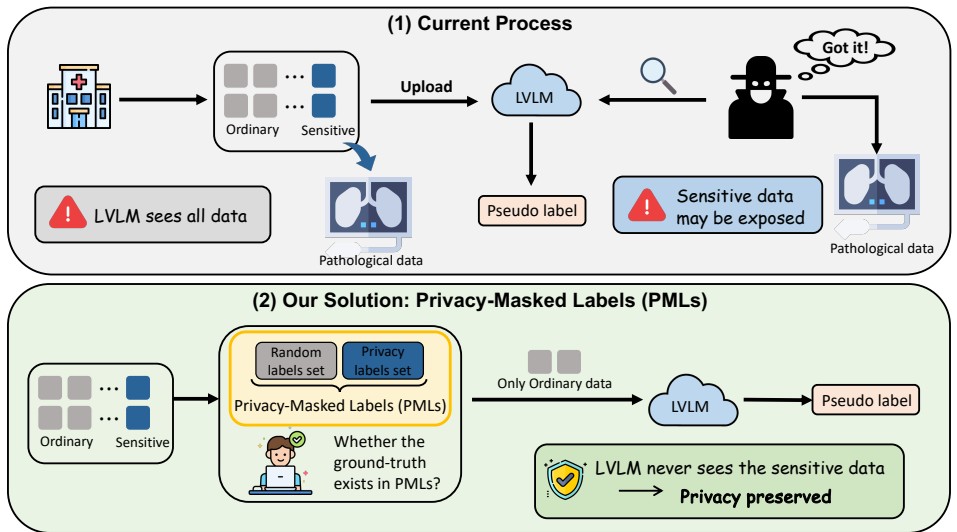

Figure 1: A comparison between current pseudo-labeling process and our PMLs labeling setting.

ically, as shown in Figure 1 (b), the PMLs merge two components: (1) a fixed set of privacy-sensitive labels; (2) a randomly sampled subset of non-privacy labels to mask the presence of sensitive labels. Human annotators are shown only the PMLs and asked to verify whether the ground-truth label is included. If it is not, the sample is then assigned to the LVLMs for pseudo-label generation. It is worth noting that any sample with a privacy-sensitive label will only be labeled by a human. This setting prevents LVLMs from directly accessing privacy-sensitive samples, thereby enhancing privacy protection. Furthermore, by introducing random sets, PMLs offer partial supervision for non-privacy samples, which effectively suppresses the noise inherent in LVLM-generated pseudo labels.

To effectively leverage the PMLs data, we theoretically derive a risk-consistent estimator to ensure the statistical consistency of the empirical risk minimization process under the condition that LVLMs exclude sensitive categories. Furthermore, we propose a hybrid probability estimation method that integrates the label probability distribution produced by LVLMs (after excluding sensitive labels) with the outputs of the training classifier, thereby improving the accuracy of conditional probability estimation. Extensive experimental results on multiple benchmark datasets validate the effectiveness of the proposed method. Our results highlight the overlooked privacy risks in pseudo-label pipelines and demonstrate a practical solution for privacy-preserving large-scale image classification. Our contributions can be summarized as follows:

- We highlight an overlooked privacy risk in LVLMs, showing that conventional training and pseudo-labeling pipelines can expose sensitive categories to the LVLMs, which may memorize and leak such private information during downstream tasks.

- We propose a novel privacy-aware PMLs setting, which integrates a fixed sensitive label set with a randomly sampled non-sensitive subset. This design ensures that sensitive classes are never exposed to the LVLMs, while the randomization masks the presence of sensitive classes and enhances robustness.

- We introduce a risk-consistent estimator that enables effective learning from the PMLs. Extensive experiments on benchmark datasets demonstrate the effectiveness of our method.

## 2 METHODOLOGY

In this section, we focus on learning from Privacy-Masked Labels (PMLs). We begin by introducing the problem setting and the labeling process of PMLs. Building on this foundation, we develop a risk-consistent estimator to effectively learn from these PMLs data.

## 2.1 PRELIMINARIES

**Ordinary Multi-Class Classification.** Let $\mathcal{X} \subset \mathbb{R}^d$ represents the $d$-dimensional feature space and $\mathcal{Y} = \{1, 2, \ldots, K\}$ denotes the label space, where $K$ is the size of the label space. For an instance $x$, the ground-truth label is denoted by $y$. Each sample $(x, y)$ is drawn from an unknown probability distribution with density $p(x, y)$. The goal of ordinary multi-class classification is to train a classifier $f(x) \colon \mathbb{R}^d \to \mathbb{R}^K$ that minimizes the classification risk:

$$R(f) = \mathbb{E}_{(x,y) \sim p(x,y)} \mathcal{L}(f(x), y), \tag{1}$$

where $\mathbb{E}_{(x,y) \sim p(x,y)}$ denotes the expectation over density $p(x, y)$ and $\mathcal{L} \colon \mathbb{R}^K \times \mathcal{Y} \to \mathbb{R}$ refers to the multi-class classification loss function.

**Pseudo Labels from LVLMs.** LVLMs are widely used to generate pseudo labels for unlabeled data (Sun et al., 2022; Xing et al., 2024; Zhang et al., 2024; Hu et al., 2024). In a typical pipeline, an image $x$ is first uploaded to an external server where the LVLMs reside. The model then predicts a label distribution across the full label space $\mathcal{Y}$, and the most confident prediction is used as the pseudo label $\hat{y}$ (Xing et al., 2024; Zhang et al., 2024). These pseudo labels are subsequently treated as ground-truth in training, allowing the learner to leverage large-scale unlabeled datasets with reduced annotation cost. Although this approach has been shown to improve classification performance, it comes with important privacy concerns. Since the generation of pseudo-labels requires uploading each image to the LVLMs, all data, including privacy-sensitive information, is directly exposed to the model provider. As a result, there is a significant risk that the LVLMs provider could misuse the data, potentially compromising user privacy.

## 2.2 PRIVACY-MASKED LABELS

The above observations motivate us to investigate a new setting that prevents LVLMs from accessing privacy-sensitive data while reducing the annotation costs in ordinary multi-class classification. In this paper, we propose a labeling framework called Privacy-Masked Labels (PMLs), which ensures that privacy-sensitive information remains hidden from LVLMs during pseudo labeling while still reducing annotation cost on non-privacy data.

**Problem Formalization of PMLs.** We consider a scenario where a subset of the label space $\mathcal{Y}$ corresponds to privacy-sensitive categories. Define the set of privacy labels as $Y_{pl} = \{pl_1, pl_2, \ldots, pl_m\} \subset \mathcal{Y}$, and the set of non-privacy labels as $Y_{npl} = \mathcal{Y} \setminus Y_{pl}$, where $m$ denotes the number of privacy-sensitive labels. From $Y_{npl}$, we sample a small subset of non-privacy labels, denoted as $Y_{rl} = \{rl_1, rl_2, \ldots, rl_r\}$, where $r$ denotes the size of $Y_{rl}$ and $r \ll K - m$. For each instance, we then construct a candidate label set $Y = Y_{pl} \cup Y_{rl}$. Let $D = \{(x_i, Y_i, S_i)\}_{i=1}^{N}$ be sampled independently from an unknown probability distribution with density $p(x, Y, S)$, where $N$ denotes the number of training samples. Here, $S$ is an indicator variable defined as

$$S = \begin{cases} 0, & \text{if } y \in Y, \\ 1, & \text{if } y \notin Y, \end{cases} \tag{2}$$

where $y$ is the ground-truth label. For samples with $S = 0$, human annotators provide the ground-truth label. For samples with $S = 1$, the LVLM is permitted to generate a pseudo label. This setting alleviates practical constraints: human annotation is used only when privacy-sensitive labels are involved, while pseudo-labeling reduces annotation costs for non-sensitive cases, without exposing privacy-sensitive samples to the LVLM.

**Superiority of PMLs.** The PMLs offer several key advantages. First, PMLs substantially reduce human annotation costs. By leveraging the strong generalization capability of large vision-language models (LVLMs) to generate relatively reliable pseudo-labels for non-privacy samples, they minimize the manual effort required in traditional classification tasks. Second, PMLs effectively prevent LVLMs from accessing privacy-sensitive data. Throughout the entire process, the LVLM never interacts with training data containing privacy-sensitive labels, thereby mitigating the risk of privacy leakage. Third, the introduction of a randomized label set provides partial supervision for non-privacy samples, which helps reduce the noise inherent in pseudo-label generation by LVLMs.

Figure 2: The architecture of our framework.

## 2.3 RISK-CONSISTENT ESTIMATOR

In this section, we introduce a risk-consistent estimator that allows us to approximate the true classification risk using the PMLs labeled data. Our goal is to learn a multi-classifier $f(x)$ from these PMLs labeled data that minimizes the expected risk in Eq. (1). In this novel setting, $S$ is indicated based on whether the ground-truth $y$ exists in the provided $Y$. Given the candidate set $Y$ and indicator $S$, we can rewrite the conditional probability of $P(y = j|x)$ by the following lemma.

**Lemma 1.** *For any instance $x$, given the candidate labels set $Y$ and the indicator variable $S$, the conditional probability $P(y = j|x)$ can be rewritten as*

$$P(y = j|x) = \sum_Y P(y = j|Y, S = 0, x)P(Y, S = 0|x)$$
$$+ \sum_Y P(y = j|Y, S = 1, x)P(Y, S = 1|x). \quad (3)$$

The proof is provided in Appendix B.1. Instead of relying on a single observed label, we compute the conditional probability of each label given $x$ and $Y$. This formulation ensures that minimizing the estimated risk is consistent with minimizing the classification risk in Eq. (1) under the privacy masking constraint (Mohri et al., 2018; Feng et al., 2020; Xu et al., 2022). Based on Lemma 1, a risk-consistent estimator for learning from PMLs can be derived by the following theorem.

**Theorem 2.** *The classification risk in Eq. (1) can be expressed as*

$$R(f) = \mathbb{E}_{(x,Y,S)\sim p(x,Y,S=0)} \sum_{j \in Y} P(y = j|Y, S = 0, x)\mathcal{L}(f(x), j)$$
$$+ \mathbb{E}_{(x,Y,S)\sim p(x,Y,S=1)} \sum_{j \notin Y} P(y = j|Y, S = 1, x)\mathcal{L}(f(x), j) \quad (4)$$
$$= \mathbb{E}_{(x,y)\sim p(x,y)}\mathcal{L}(f(x), y) + \mathbb{E}_{(x,Y,S)\sim p(x,Y,S=1)} \sum_{j \notin Y} P(y = j|Y, S = 1, x)\mathcal{L}(f(x), j)$$
$$= R_{PML}(f),$$

*where $R_{PML}(f)$ denotes the classification risk of learning from PML-labeled data.* The proof is provided in Appendix B.2. This equality holds because $P(y|Y, S, x)$ integrates to the true posterior distribution of $y$ given $x$, $Y$ and $S$. Thus the expectation risk of $R_{PML}(f)$ coincides with classification risk in Eq. (1).

**Remark 3.** *Since the training dataset $D = \{(x_i, Y_i, S_i)\}_{i=1}^N$ is sampled independently from the $p(x, Y, S)$, the empirical risk estimator can be naively approximated as*

$$\widehat{R}_{PML}(f) = \frac{1}{N_{S=0}} \sum_{i=1}^{N_{S=0}} \mathcal{L}(f(x_i), y) + \frac{1}{N_{S=1}} \sum_{i=1}^{N_{S=1}} \sum_{j \notin Y} P(y = j|Y, S = 1, x)\mathcal{L}(f(x), j), \quad (5)$$

where $N_{S=0}$ and $N_{S=1}$ denote the number of samples with $S = 0$ and $S = 1$, respectively. Then, we can learn a multi-class classifier $f(x)$ by minimizing the proposed empirical approximation of the risk-consistent estimator in Eq. (4). The estimator has two desirable consequences. First, it allows us to train with supervision without introducing bias for the samples with $S = 0$. Second, it ensures that the LVLM can be used to generate pseudo labels for non-privacy samples, thereby reducing annotation cost.

**Conditional Probability Estimation.** In practice, the conditional probability $P(y = j|Y, S = 1, x)$ is generally hard to estimate directly. To alleviate this, we estimate this conditional probability via a convex combination of LVLM-generated probabilities and the classifier's own softmax outputs:

$$\widehat{P}(y = j|Y, S = 1, x) = \lambda \cdot \sigma(f_j(x)) + (1 - \lambda) \cdot \pi_{\mathbf{LVLM}}(y = j|x, \mathcal{Y} \setminus Y), \tag{6}$$

where $\lambda \in [0, 1]$ is a balancing coefficient, $\sigma(f_j(x)) = Softmax(f_j(x)/\tau)$ is the normalized output of j-th classifier, and $\pi_{\mathrm{LVLM}}(\cdot)$ denotes the posterior probability distribution generated from LVLM. Here, we only require the LVLM to provide the probability distribution over $\mathcal{Y} \setminus Y$. To this end, we employ an LVLM (e.g., CLIP (Radford et al., 2021)) to compute the cosine similarity between the image embedding and the textual embedding for each class in $\mathcal{Y} \setminus Y$, thereby constructing a probability distribution over $\mathcal{Y} \setminus Y$. This combination integrates complementary strengths, with the LVLM providing semantic priors and the classifier adapting to domain distributions, while the weighting mechanism balances overconfidence and noise, producing high-quality label distributions (which can be validated in Section 3.3). Using this formulation, the conditional probability $P(y = j|Y, S = 1, x)$ is estimated as $\widehat{P}(y = j|Y, S = 1, x)$, which enables the computation of an empirical risk-consistent objective to optimize the classifier $f(s)$ based on the estimated probability.

### 2.4 PRACTICAL IMPLEMENTATION

**Model.** Our model consists of a frozen backbone feature extractor (e.g., ViT-B/32-based CLIP) and a trainable LaFTer (Mirza et al., 2023) adapter head $f(x)$ producing class logits. The adapter enables efficient fine-tuning under PMLs constraints, while the frozen backbone preserves general visual representations. For a comprehensive understanding, Figure 2 illustrate the training procedure under the PMLs setting. This ensures that privacy-sensitive samples rely exclusively on human labels, while non-sensitive cases exploit LVLM guidance. The adaptive mixture of probabilities stabilizes training and reduces noise from pseudo-labels.

**Loss Functions.** A variety of loss functions are compatible with our framework, including logistic loss $\mathcal{L}(f(x), y) = \log(1 + e^{-yf(x)})$ and mean-squared error loss $\mathcal{L}(f(x), y) = (f(x) - y)^2$. For the experiments presented in this work, we employ the cross-entropy loss, which is widely regarded as the standard choice for multi-class classification and provides stable optimization performance.

## 3 EXPERIMENTS

### 3.1 EXPERIMENTAL SETUP

**Dataset.** To assess the performance of the proposed method, we conduct comprehensive experiments on eight widely used multi-class classification datasets. These datasets span three major domains: natural object recognition (Caltech-101 (Fei-Fei et al., 2004), CIFAR-100 (Krizhevsky et al., 2009), Oxford_Pets (Parkhi et al., 2012), DTD (Cimpoi et al., 2014)), fine-grained category classification (Food-101 (Bossard et al.), Stanford Cars (Krause et al., 2013), Flowers-102 (Nilsback & Zisserman, 2008)), and action recognition (UCF-101 (Soomro et al., 2012)). During training, the original labels of all images are replaced with Privacy-Masked Labels (PMLs), whereas the test sets retain their ground-truth labels to ensure fair evaluation.

**Implementation Details.** For a fair and consistent comparison across experiments, we adopt the same vision backbone (i.e., ViT-B/32-based CLIP (Radford et al., 2021)) and optimization strategy. The trainable LaFTer adapter head classifier $f(x)$ is optimized with AdamW using an initial learning rate of $5e^{-4}$ and weight decay of $1e^{-4}$. We train all models for 50 epochs with a batch size of 50 on a single NVIDIA RTX 4090 GPU. To examine the impact of random labels set size, we evaluate each dataset under several random label ratios. Since the number of categories differs across datasets, we use a unified metric to measure the effectiveness of Privacy-Masked Labels (PMLs). Let $q = \frac{r}{K-m}$

Table 1: Test accuracy (%) for using CLIP-generated PMLs. The best method is highlighted in **bold** and the second-best method is underlined.

| | | CIFAR-100 | Food-101 | Caltech-101 | Oxford Pets | DTD | Flowers-102 | Stanford Cars | UCF-101 | Average |
|---|---|---|---|---|---|---|---|---|---|---|
| | Fully supervised | 84.41 | 84.99 | 96.71 | 91.88 | 71.45 | 98.54 | 81.43 | 85.30 | 86.84 |
| | Zero-shot CLIP (Radford et al., 2021) | 63.70 | 80.15 | 87.90 | 84.30 | 43.48 | 66.70 | 59.27 | 64.50 | 68.75 |
| Partial-label | DIRK (Wu et al., 2024) | 76.62 | 72.50 | 92.17 | 81.39 | 47.81 | 69.23 | 59.72 | 64.68 | 70.52 |
| | PaPi (Xia et al., 2023) | 79.05 | 75.23 | 91.60 | 82.69 | 47.22 | 68.78 | 60.08 | 66.85 | 71.44 |
| | SPMI (Liu et al., 2024) | 59.39 | 60.01 | 85.72 | 74.35 | 42.55 | 59.07 | 36.66 | 52.82 | 58.82 |
| Com-label | PLNL (Li et al., 2025a) | 77.19 | 71.54 | 93.02 | 82.07 | 50.12 | 70.85 | 59.60 | 65.32 | 71.21 |
| Pseudo-label | CPL_RC (Zhang et al., 2024) | 70.83 | 78.78 | 90.67 | 85.34 | 51.89 | 71.54 | 58.34 | 64.71 | 71.51 |
| | CPL_CC (Zhang et al., 2024) | 75.36 | 80.69 | 93.38 | 88.28 | 51.95 | 73.61 | 60.48 | 67.46 | 73.90 |
| | CPL_LW (Zhang et al., 2024) | 70.85 | 78.95 | 91.24 | 85.31 | 52.19 | 71.70 | 58.50 | 64.79 | 71.69 |
| | LaFTer (Mirza et al., 2023) | 74.45 | 80.09 | 93.02 | 85.39 | 50.23 | 70.65 | 54.97 | 65.64 | 71.81 |
| | **PMLL (Our)** | **80.75** | **82.36** | **95.34** | **90.65** | **60.11** | **83.03** | **71.17** | **75.28** | **79.84** |

denotes the ratio of random labels to all non-privacy labels, where $r$ denotes the number of random labels, $m$ denotes the count of privacy-sensitive labels, and $K$ denotes the total number of classes. This metric normalizes the degree of label randomization relative to the available non-privacy label space. In the results section, we report model performance under different $q$ values and varying amounts of privacy-sensitive labels to analyze robustness.

**Compared Methods.** To validate the effectiveness of our method, we first compared it with CLIP Zero-shot (Radford et al., 2021) and label-free LaFTer (Mirza et al., 2023). Furthermore, we compare our method with recent weakly supervised learning method, including partial-label learning (PaPi (Xia et al., 2023), DIRK (Wu et al., 2024), SPMI (Liu et al., 2024)), complementary-label learning (PLNL (Li et al., 2025a)), and LVLM-pseudolabel learning (CPL (Zhang et al., 2024)).

## 3.2 MAIN RESULTS

**Using VL-Contrastive-generated PMLs.** Table 1 presents the comparative performance of using the Vision-Language Contrastive Model, CLIP (Radford et al., 2021), to generate PMLs. It compares the proposed PMLL with recent partial-label, complementary-label, and pseudo-label learning methods across eight widely used benchmark datasets. PMLL consistently achieves state-of-the-art performance, surpassing all compared methods by a considerable margin. Notably, PMLL achieves an improvement of 5.94% over the strongest baseline CPL_CC (Zhang et al., 2024). These results validate the effectiveness of the proposed PMLL, establishing a strong new benchmark and paving the way for future research on learning with LVLM-generated labels.

**Using QA-LVLM-generated PMLs.** Table 2 reports the performance of using PMLs generated by Qwen (Question-Answer LVLM). Consistent with the CLIP-based results in Table 1, PMLL achieves state-of-the-art accuracy across all datasets. A comparison with Table 1 leads to three key observations: ① Under identical experimental conditions, the proposed PMLL consistently outperforms the recent weakly supervised and pseudo-labeling baselines, underscoring its effectiveness. ② Qwen proves to be a strong generator of PMLs: nearly all methods yield higher performance when using Qwen-generated PMLs compared to those produced by CLIP. This result suggests that more capable and general LVLMs can produce higher-quality PMLs, thereby offering more reliable supervisory signals for downstream classification tasks. Notably, PMLL achieves the best overall performance in this setting. ③ When using Qwen to generate PMLs, PMLL approaches the accuracy of fully supervised baselines. This finding indicates that PMLs can effectively exploit increasingly powerful LVLMs and, in the longer term, have the potential to significantly reduce annotation costs while narrowing the gap with fully supervised learning.

## 3.3 FURTHER ANALYSES

**Effectiveness of Pseudo-label-based RC Estimator.** Table 3 shows the effect of adding true labels, pseudo labels, and risk-consistent (RC) estimator. Using only pseudo labels (setup (iii)) leads to a marked accuracy drop due to distribution shift and label noise. Introducing a small set of ground-truth labels (setup (ii)) stabilizes training and improves accuracy, and combining them with pseudo

Table 2: Test accuracy (%) for using Qwen-generated PMLs. The best method is highlighted in **bold** and the second-best method is underlined.

| | | CIFAR-100 | Food-101 | Caltech-101 | Oxford Pets | DTD | Flowers-102 | Stanford Cars | UCF-101 | Average |
|---|---|---|---|---|---|---|---|---|---|---|
| | Fully supervised | 84.41 | 84.99 | 96.71 | 91.88 | 71.45 | 98.54 | 81.43 | 85.30 | 86.84 |
| | Zero-shot Qwen (Bai et al., 2025) | 67.48 | 80.39 | 89.36 | 86.17 | 57.62 | 70.73 | 69.32 | 70.10 | 73.89 |
| Partial-label | DIRK (Wu et al., 2024) | 77.84 | 72.55 | 93.35 | 80.29 | 56.32 | 76.29 | 70.02 | 74.10 | 75.10 |
| | PaPi (Xia et al., 2023) | 80.21 | 75.04 | 92.86 | 80.59 | 56.62 | 75.15 | 70.87 | 76.21 | 75.94 |
| | SPMI (Liu et al., 2024) | 60.81 | 58.96 | 87.26 | 71.55 | 51.01 | 65.94 | 44.67 | 63.76 | 63.00 |
| Com-label | PLNL Li et al. (2025a) | 75.98 | 70.05 | 94.12 | 80.67 | 60.28 | 83.23 | 71.79 | 74.73 | 76.36 |
| Pseudo-label | CPL_RC (Zhang et al., 2024) | 70.66 | 78.77 | 90.67 | 85.36 | 51.89 | 71.51 | 58.35 | 64.77 | 71.51 |
| | CPL_CC Zhang et al. (2024) | 75.28 | 80.69 | 93.39 | 88.12 | 51.95 | 73.67 | 60.42 | 67.33 | 73.87 |
| | CPL_LW (Zhang et al., 2024) | 70.87 | 79.98 | 90.83 | 85.72 | 52.18 | 73.17 | 58.54 | 64.83 | 72.02 |
| | LaFTer | 74.45 | 80.09 | 93.02 | 85.39 | 50.23 | 70.65 | 54.97 | 65.64 | 71.81 |
| | **PMLL (Our)** | **81.19** | **82.65** | **95.78** | **89.26** | **66.31** | **87.05** | **76.68** | **81.92** | **82.61** |

Table 3: Effect of adding true labels (S=0), pseudo-labels (S=1) and risk-consistent (RC) estimator.

| | True-label (S=0) | Pseudo-label (S=1) | RC | CIFAR-100 | Food-101 | Caltech-101 | Oxford Pets | DTD | Flowers-102 | Stanford Cars | UCF-101 | Average |
|---|---|---|---|---|---|---|---|---|---|---|---|---|
| (i) | ✗ | ✗ | ✗ | 74.45 | 80.09 | 93.02 | 85.39 | 50.23 | 70.65 | 54.97 | 65.64 | 71.81 |
| (ii) | ✓ | ✗ | ✗ | 76.74 (+2.29) | 80.91 (+0.82) | 93.51 (+0.49) | 85.91 (+0.52) | 59.77 (+9.54) | 80.24 (+9.59) | 61.63 (+6.66) | 68.24 (+2.60) | 75.87 (+4.06) |
| (iii) | ✗ | ✓ | ✗ | 71.34 (-5.40) | 75.13 (-5.78) | 92.45 (-1.06) | 81.52 (-4.39) | 46.34 (-13.43) | 69.79 (-10.45) | 49.99 (-11.64) | 62.78 (-5.46) | 68.67 (-7.20) |
| (iV) | ✓ | ✓ | ✗ | 77.19 (+5.85) | 80.51 (+5.38) | 92.99 (+0.54) | 86.59 (+5.07) | 58.33 (+11.99) | 80.97 (+11.18) | 63.50 (+13.51) | 70.68 (+7.90) | 76.97 (+8.30) |
| (V) | ✓ | ✓ | ✓ | **80.75** (+3.56) | **82.36** (+1.85) | **95.34** (+2.35) | **90.65** (+4.06) | **60.11** (+1.78) | **83.03** (+2.06) | **71.17** (+7.67) | **75.28** (+4.60) | **79.84** (+2.87) |
| Total ↑ (Compared to (i)) | | | | (+6.30) | (+2.27) | (+2.32) | (+5.26) | (+9.88) | (+12.38) | (+16.20) | (+9.64) | (+8.03) |

labels (setup (iv)) brings further gains by exploiting unlabeled data. Our full method (setup (V)) achieves the best performance across all datasets. The RC estimator plays a crucial role in aligning pseudo labels with the true label distribution, leading to substantial performance gains, especially on fine-grained datasets. These results highlight the effectiveness of our framework.

**Impact of Random Set Ratio** $q$**.** Figure 3 reports the average accuracy on six datasets of all methods under different labeled ratios (Detailed results for each dataset, along with additional experiments, are provided in Appendix C.1). The size of the random set is crucial in our setting. When the random set is too small, the cost of manual annotation can be greatly reduced, but the supervisory signals for non-private samples become extremely scarce, which negatively impacts the model's performance. To investigate the effect of the random set size, we conduct an ablation study with varying random set ratio. As the ratio increases from 0.05 to 0.3, PMLL improves steadily and remains the best across all settings. Remarkably, even with the smallest ratio of 0.05, our method outperforms the strongest competitors by a large margin, demonstrating its robustness and efficiency under extremely limited annotation budgets. This highlights that our approach achieves high accuracy while effectively mitigating the trade-off between privacy risk and annotation cost.

**Privacy-sensitive Category Proportion Study.** In practical scenarios, multiple classes may be privacy-sensitive. To assess the influence of the number of privacy-sensitive classes, we perform an ablation study for multiple privacy classes. Figure 4 shows the average accuracy six benchmark datasets with varying numbers of privacy-sensitive classes (Detailed results for each dataset are provided in Appendix C.2). It is evident that the proposed PMLL consistently achieves the highest average accuracy across all privacy-sensitive class sizes. Additionally, PMLL maintains stable performance as the privacy class size increases, without noticeable degradation, which demonstrates that PMLL is robust to the number of privacy-sensitive classes.

**Accuracy on privacy-sensitive classes** $\mathcal{Y}_{pl}$**.** Figure 5 presents the classification accuracy on the privacy-sensitive categories $\mathcal{Y}_{pl}$ (Additional experiments are provided in Appendix C.3). Across all four datasets, the proposed PMLL achieves consistently superior performance compared to all baselines. In particular, our method attains an average accuracy of 86.25, yielding a substantial improvement over the strongest compared method. These results clearly demonstrate the effectiveness of the proposed PMLL in accurately recognizing privacy-sensitive categories.

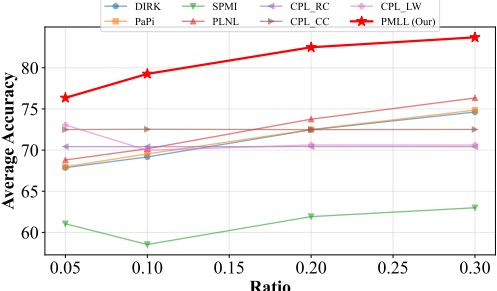

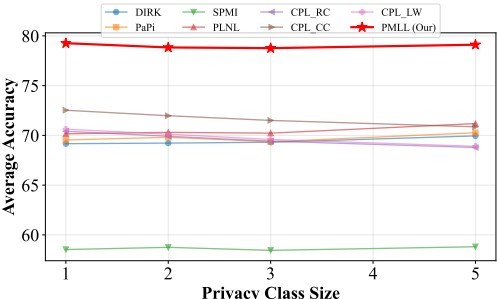

Figure 3: Impact of random set ratio $q$ on accuracy. Experiments are performed on CLIP-generated PMLs data.

Figure 4: Influence of varying numbers of privacy-sensitive categories. Experiments are performed on CLIP-generated PMLs data.

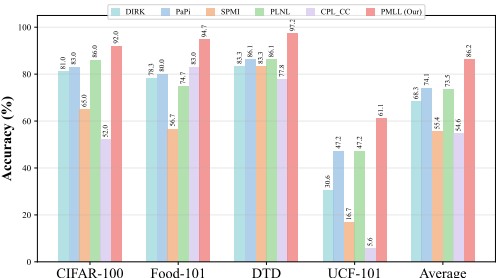

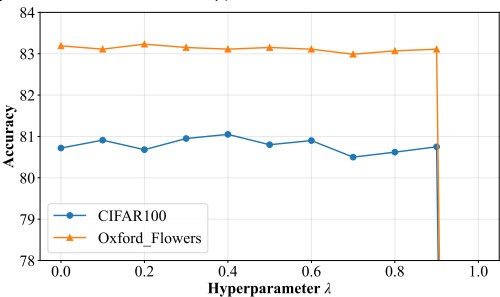

Figure 5: Accuracy on privacy-sensitive classes. Experiments are performed on CLIP-generated PMLs data.

Figure 6: Influence of hyperparameter $\lambda$. Experiments are performed on CLIP-generated PMLs data.

**Influence of Hyperparameter $\lambda$.** In Figure 6, we investigate the impact of the conditional probability hyperparameter $\lambda$ on both a coarse-grained dataset (CIFAR-100) and a fine-grained dataset (Flowers-102). As $\lambda$ increases from 0 to 1, the overall performance exhibits a clear downward trend. Nevertheless, our method maintains relatively stable accuracy when $\lambda$ lies within the range $[0, 0.9]$. This observation suggests that PMLL is capable of adaptively leveraging the interaction between the LVLM and the learned classifier to estimate the conditional probability distribution effectively.

**Comparison of Training Cost.** In this section, we compare the training cost of the proposed PMLL with various compared methods. The experiments utilize the CIFAR-100, Caltech-101 and UCF-101 datasets, conducted on a single NVIDIA RTX 4090 GPU, with

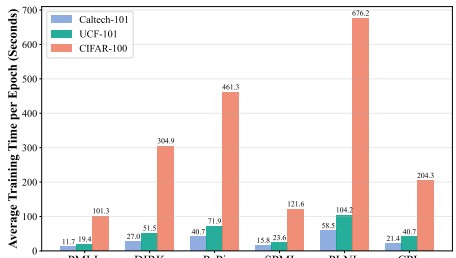

Figure 7: Comparison of training cost between PMLL and compared methods. The numbers in the figure represent the average time (in seconds) required to train each method for single epoch.

all other settings consistent with those of the previous experiments. As shown in Figure 7, we measure the time (in seconds) required to train each method for one epoch. PMLL significantly reduces training time compared to other compared methods, which demonstrate that PMLL achieves higher accuracy while requiring less training time. (Additional visualizations comparing all training epochs are provided in the Appendix C.4)

## 4 RELATED WORK

### 4.1 PSEUDO LABELING WITH LVLMS

Large vision language models (LVLMs) are typically trained on large-scale datasets that contain paired image–text annotations or classification labels (Radford et al., 2021; Dai et al., 2023; Liu et al., 2023; Peng et al., 2023; Ormazabal et al., 2024; Zhu et al., 2024). They have demonstrated

strong generalization ability across a wide range of downstream tasks, including image classification and caption generation (Deitke et al., 2025; Lee et al., 2025; Li et al., 2025b; Bai et al., 2025). To reduce the high cost of manual annotation in traditional classification datasets, recent studies have explored the use of LVLMs to generate pseudo labels as an alternative to supervised labeling (Sun et al., 2022; Zhang et al., 2024; Hu et al., 2024). However, real-world datasets frequently include highly sensitive information—such as personal identities, medical records. Uploading such data to LVLMs offers no assurance that proprietary models will refrain from misuse, potentially compromising user privacy and raising severe ethical and security concerns. (Wang et al., 2025; Guo et al., 2025).

## 4.2 Privacy-Preserving Machine Learning

Privacy-preserving machine learning has been extensively studied, with methods such as privacy-label learning (Li et al., 2024), partial-label learning (Feng et al., 2020; Zhang et al., 2021; Xia et al., 2023; Jia et al., 2024; Liu et al., 2024; Wu et al., 2024), and complementary-label learning (Ishida et al., 2019; Chou et al., 2020; Gao & Zhang, 2021; Wei et al., 2023; Li et al., 2025a) being widely explored. Privacy-label learning is a novel privacy-preserving setting that aims to protect sensitive labels during the annotation process (Li et al., 2024). However, this setting cannot leverage pseudo-labels generated by LVLMs and instead relies strictly on human-annotated labels. Partial-label learning and complementary-label learning are two widely adopted privacy-preserving settings (Ishida et al., 2019; Zhang et al., 2021). In partial-label learning, each instance is associated with a set of candidate labels (which may contain noisy labels) (Jia et al., 2024), whereas in complementary-label learning, each instance is provided with a label indicating a class that it does not belong to (Wei et al., 2023; Li et al., 2025a). While these techniques mitigate certain privacy concerns, they primarily focus on protecting data samples rather than the labels themselves. In contrast, our work addresses a complementary and underexplored dimension: preventing privacy-sensitive data from ever being exposed to LVLMs, offering a practical alternative to learning from LVLM-generated pseudo-labels.

## 5 Conclusion

The pervasive use of Large Vision-Language Models (LVLMs) for pseudo-labeling comes with a hidden cost: the involuntary exposure of sensitive user data to proprietary models. In this work, we confront this challenge head-on by introducing Privacy-Masked Labels (PMLs), a novel framework that prevents LVLMs from accessing privacy-sensitive data during annotation. By integrating a fixed set of privacy-sensitive labels with a randomized non-privacy subset, PMLs ensure that images belonging to privacy-sensitive categories (e.g., medical conditions, personal identities) are never processed by external LVLMs. Coupled with our risk-consistent estimator, which intelligently leverages LVLM-generated pseudo-labels for non-privacy data, our method achieves a best-of-both-worlds outcome: it significantly reduces annotation costs while providing privacy guarantees.

The implications of our work extend beyond academic benchmarks. PMLs offer a practical and immediately applicable pathway for industries handling sensitive visual data to safely leverage powerful LVLMs without compromising user privacy or violating evolving data regulations (e.g., GDPR (Kuner et al., 2021)). For instance: **(1) Healthcare AI:** Medical institutions can utilize public LVLMs to annotate non-privacy medical images (e.g., common anatomy) while ensuring that images revealing rare diseases or patient identities are kept entirely in-house. **(2) Smart Surveillance:** Security systems can classify public behavior patterns using LVLMs without exposing footage of private spaces or identifiable individuals to third-party models.

By bridging the gap between data utility and privacy preservation, our framework provides an effective solution that utilizes LVLM to generate pseudo labels. We envision a future where LVLM is not only efficient but also ethically grounded. We believe this work marks a critical step toward that goal, and we open-source our code to encourage widespread adoption and further innovation in secure, privacy-aware machine learning.

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

## A    THE USE OF LARGE LANGUAGE MODELS (LLMS)

During the preparation of this manuscript, large language models (e.g., ChatGPT) were used solely for grammar checking, language polishing and enhancing readability. All initial drafts of the manuscript were written entirely by the authors. The authors carefully reviewed all AI-generated suggestions to ensure accuracy and academic rigor.

## B    THEORETICAL PROOFS

### B.1    PROOF FOR LEMMA 1

**Lemma 1.** *For any instance $x$, given the candidate labels set $Y$ and the indicator variable $S$, the conditional probability $P(y = j|x)$ can be rewritten as*

$$P(y = j|x) = \sum_Y P(y = j|Y, S = 0, x)P(Y, S = 0|x)$$
$$+ \sum_Y P(y = j|Y, S = 1, x)P(Y, S = 1|x).$$

*Proof.* Suppose $S$ indicated whether the ground-truth $y$ exists in the provided labels set $Y$. Given the candidate labels set $Y$ and the indicator $S$, we can rewrite the conditional probability $P(y = j \mid x)$ as follows:

$$P(y = j|x) = \sum_Y P(y = j, Y|x)$$
$$= \sum_{z=0}^{1} \sum_Y P(y = j, Y, S = z|x)$$
$$= \sum_Y P(y = j, Y, S = 0|x) + \sum_Y P(y = j, Y, S = 1|x)$$
$$= \sum_Y P(y = j|Y, S = 0, x)P(Y, S = 0|x) + \sum_Y P(y = j|Y, S = 1, x)P(Y, S = 1|x),$$
(7)

which concludes the proof of Lemma 1. $\qquad\square$

### B.2    PROOF FOR THEOREM 2

**Theorem 2.** *The classification risk in Eq. ([1](#)) can be expressed as*

$$R(f) = \mathbb{E}_{(x,Y,S)\sim p(x,Y,S=0)} \sum_{j\in Y} P(y = j|Y, S = 0, x)\mathcal{L}(f(x), j)$$
$$+ \mathbb{E}_{(x,Y,S)\sim p(x,Y,S=1)} \sum_{j\notin Y} P(y = j|Y, S = 1, x)\mathcal{L}(f(x), j)$$
$$= \mathbb{E}_{(x,y)\sim p(x,y)}\mathcal{L}(f(x), y) + \mathbb{E}_{(x,Y,S)\sim p(x,Y,S=1)} \sum_{j\notin Y} P(y = j|Y, S = 1, x)\mathcal{L}(f(x), j)$$
$$= R_{PML}(f),$$

*where $R_{PML}(f)$ denotes the classification risk of learning from PML-labeled data.*

*Proof.* By using Lemma 1, the classification risk in Eq. (1) can be expressed as

$$R(f) = \mathbb{E}_{(x,y) \, p(x,y)} \mathcal{L}(f(x), y)$$

$$= \mathbb{E}_X \sum_{j=1}^{K} P(y = j|x) \mathcal{L}(f(x), j)$$

$$= \mathbb{E}_X \sum_{j=1}^{K} \sum_{Y} P(y = j, Y|x) \mathcal{L}(f(x), j)$$

$$= \mathbb{E}_X \sum_{j=1}^{K} \left\{ \sum_{Y} P(y = j|Y, S = 0, x) P(Y, S = 0|x) \right.$$

$$\left. + \sum_{Y} P(y = j|Y, S = 1, x) P(Y, S = 1|x) \right\} \mathcal{L}(f(x), j)$$

$$= \mathbb{E}_X \sum_{j=1}^{K} \sum_{Y} P(y = j|Y, S = 0, x) P(Y, S = 0|x) \mathcal{L}(f(x), j)$$

$$+ \mathbb{E}_X \sum_{j=1}^{K} \sum_{Y} P(y = j|Y, S = 1, x) P(Y, S = 1|x) \mathcal{L}(f(x), j)$$

$$= \mathbb{E}_{(x,S=0,Y)} \sum_{j \in Y} P(y = j|Y, S = 0, x) \mathcal{L}(f(x), j)$$

$$+ \mathbb{E}_{(x,S=1,Y)} \sum_{j \notin Y} P(y = j|Y, S = 1, x) \mathcal{L}(f(x), j)$$

$$= \mathbb{E}_{(x,y) \sim p(x,y)} \mathcal{L}(f(x), y) + \mathbb{E}_{(x,Y,S) \sim p(x,Y,S=1)} \sum_{j \notin Y} P(y = j|Y, S = 1, x) \mathcal{L}(f(x), j)$$

$$(\because \textit{For sample with } S = 0, \textit{ the ground truth is provided by human annotator.})$$

$$= R_{PML}(f),$$

$$(8)$$

which concludes the proof of Theorem 2. $\qquad\square$

## C  ADDITIONAL ANALYSIS

### C.1  ADDITIONAL EXPERIMENTS ON DIFFERENT RANDOM SET RATIOS $q$

Figure 8 and Table 5 compare our method with several baselines across different random set ratios. We observe three key trends. First, our method consistently surpasses all competitors on every dataset and ratio which shows that privacy masked labels combined with the risk consistent estimator form a highly robust framework. Second, performance steadily improves as the random set ratio increases from 0.05 to 0.3 which indicates that adding more randomized non sensitive labels not only hides the presence of sensitive categories more effectively but also provides additional context that improves pseudo label quality. Third, the largest margins appear on fine grained datasets such as Flowers 102 and Stanford Cars where the richer label space allows our model to achieve substantial accuracy gains over the strongest baseline. These results demonstrate that our approach simultaneously enhances privacy and boosts generalization and that the benefit scales favorably as more randomized context is introduced.

### C.2  ADDITION EXPERIMENTS ON VARIOUS PRIVACY-SENSITIVE CATEGORIES SIZE

Figure 9 and Table 6 report the effect of varying size on privacy sensitive category classification across multiple benchmark datasets. From these results, we can observe that the proposed PMLL method achieves the best accuracy under every privacy-sensitive categories size which indicates that the proposed privacy masked label learning framework is robust to the choice of privacy size. These

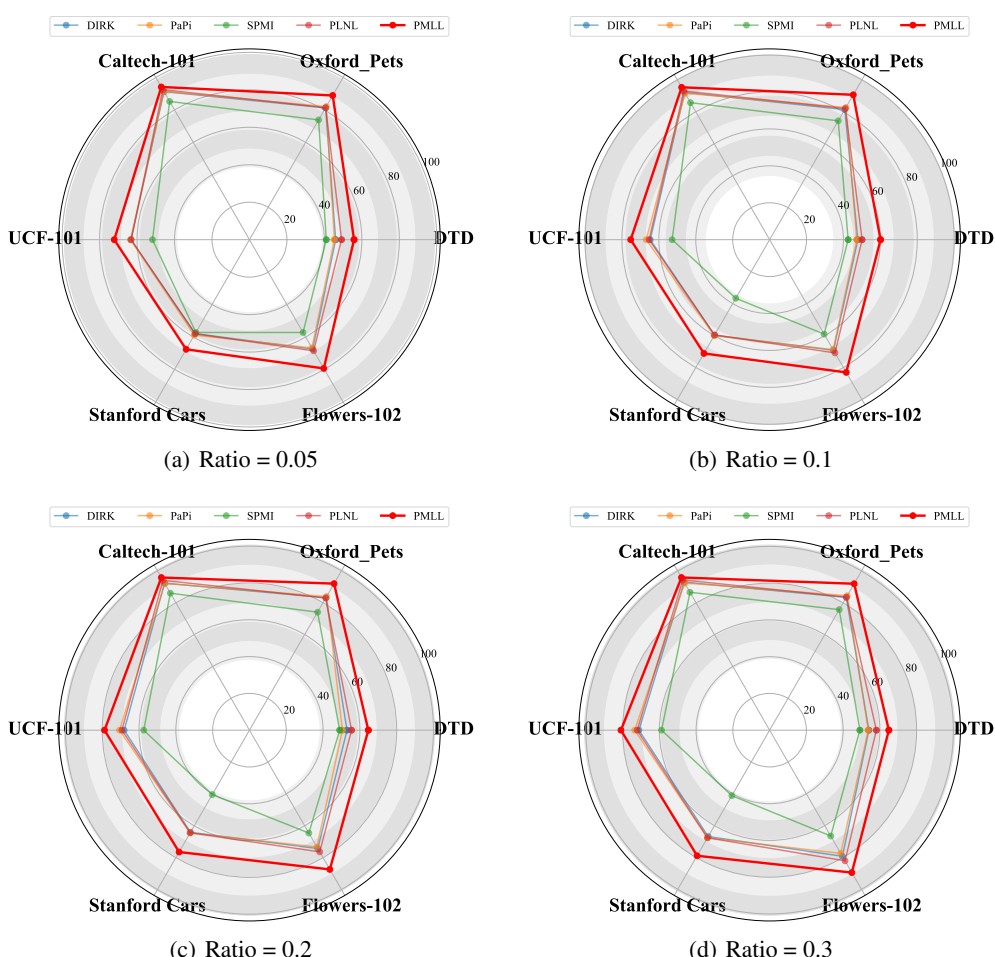

Figure 8: The classification accuracy of various methods using diverse random set ratios on six benchmark datasets. Experiments are performed on CLIP-generated PMLs data.

Table 4: Test accuracy (%) on privacy-sensitive categories. The best method is highlighted in **bold**.

|  | Method | CIFAR-100 | Food-101 | DTD | UCF-101 | Average |
|---|---|---|---|---|---|---|
| CLIP-generated PMLs | DIRK (Wu et al., 2024) | 81.00 | 78.33 | 83.33 | 30.56 | 68.31 |
|  | PaPi (Xia et al., 2023) | 83.00 | 80.00 | 86.11 | 47.22 | 74.08 |
|  | SPMI (Liu et al., 2024) | 65.00 | 56.67 | 83.33 | 16.67 | 55.42 |
|  | PLNL (Li et al., 2025a) | 86.00 | 74.67 | 86.11 | 47.22 | 73.50 |
|  | CPL (Zhang et al., 2024) | 52.00 | 83.00 | 77.78 | 5.56 | 54.58 |
|  | **PMLL (Our)** | **92.00** | **94.67** | **97.22** | **61.11** | **86.25** |
| Qwen-generated PMLs | DIRK (Wu et al., 2024) | 79.00 | 75.00 | 83.33 | 41.67 | 69.75 |
|  | PaPi (Xia et al., 2023) | 83.00 | 78.67 | 91.67 | 36.11 | 72.36 |
|  | SPMI (Liu et al., 2024) | 58.00 | 58.67 | 91.67 | 19.44 | 56.94 |
|  | PLNL (Li et al., 2025a) | 78.00 | 72.67 | 86.11 | 47.22 | 71.00 |
|  | CPL (Zhang et al., 2024) | 58.00 | 76.33 | 38.89 | 11.11 | 46.08 |
|  | **PMLL (Our)** | **91.00** | **92.33** | **97.22** | **63.89** | **86.11** |

results validate the adaptability of our approach and its ability to retain superior performance under different privacy-sensitive categories size.

Table 5: Test accuracy (%) on six benchmark datasets under different ratios. The best method is highlighted in **bold**.

| | | Caltech-101 | Oxford Pets | DTD | Flowers-102 | Stanford Cars | UCF-101 | Average |
|---|---|---|---|---|---|---|---|---|
| **Ratio = 0.05** | DIRK | 91.28 | 81.25 | 46.04 | 67.48 | 57.89 | 63.26 | 67.87 |
| | PaPi | 91.40 | 82.07 | 45.33 | 66.71 | 58.85 | 63.52 | 67.98 |
| | SPMI | 85.23 | 73.83 | 41.02 | 57.21 | 57.21 | 51.78 | 61.05 |
| | PLNL | 92.50 | 81.39 | 49.17 | 68.53 | 58.03 | 63.18 | 68.80 |
| | CPL_RC | 90.67 | 85.36 | 51.89 | 71.54 | 58.34 | 64.71 | 70.42 |
| | CPL_CC | 93.39 | 88.12 | 51.95 | 73.61 | 60.44 | 67.46 | 72.49 |
| | CPL_LW | 90.83 | 85.72 | 52.19 | 71.70 | 57.16 | 64.79 | 73.05 |
| | **PMLL (Our)** | **94.16** | **88.91** | **55.85** | **79.38** | **67.59** | **72.19** | **76.35** |
| **Ratio = 0.1** | DIRK | 92.17 | 81.39 | 47.81 | 69.23 | 59.72 | 64.68 | 69.17 |
| | PaPi | 91.60 | 82.69 | 47.22 | 68.78 | 60.08 | 66.85 | 69.54 |
| | SPMI | 85.72 | 74.35 | 42.55 | 59.07 | 36.66 | 52.82 | 58.53 |
| | PLNL | 93.02 | 82.07 | 50.12 | 70.85 | 59.60 | 65.32 | 70.16 |
| | CPL_RC | 90.67 | 85.34 | 51.89 | 71.54 | 58.34 | 64.71 | 70.42 |
| | CPL_CC | 93.38 | 88.28 | 51.95 | 73.61 | 60.48 | 67.46 | 72.53 |
| | CPL_LW | 91.24 | 85.31 | 52.19 | 71.70 | 58.50 | 64.79 | 70.62 |
| | LaFTer | 93.02 | 85.39 | 50.23 | 70.65 | 54.97 | 65.64 | 69.98 |
| | **PMLL (Our)** | **95.34** | **90.65** | **60.11** | **83.03** | **71.17** | **75.28** | **79.26** |
| **Ratio = 0.2** | DIRK | 92.25 | 83.10 | 53.01 | 74.30 | 64.08 | 68.07 | 72.47 |
| | PaPi | 92.09 | 83.87 | 50.53 | 73.29 | 64.52 | 70.61 | 72.48 |
| | SPMI | 85.96 | 74.05 | 49.05 | 64.55 | 40.42 | 57.47 | 61.92 |
| | PLNL | 94.04 | 82.91 | 55.50 | 76.37 | 64.40 | 69.31 | 73.75 |
| | CPL_RC | 90.67 | 85.36 | 51.89 | 71.54 | 58.36 | 64.71 | 70.42 |
| | CPL_CC | 93.39 | 88.12 | 51.89 | 73.61 | 60.47 | 67.46 | 72.49 |
| | CPL_LW | 90.83 | 85.72 | 52.19 | 71.70 | 58.50 | 64.79 | 70.62 |
| | LaFTer | 93.02 | 85.39 | 50.23 | 70.65 | 54.97 | 65.64 | 69.98 |
| | **PMLL (Our)** | **95.78** | **91.96** | **64.60** | **87.37** | **76.46** | **78.77** | **82.49** |
| **Ratio = 0.3** | DIRK | 93.06 | 83.43 | 54.02 | 79.25 | 66.82 | 71.13 | 74.62 |
| | PaPi | 92.45 | 84.49 | 53.66 | 77.43 | 67.67 | 73.43 | 74.86 |
| | SPMI | 86.65 | 75.74 | 49.11 | 66.50 | 41.13 | 58.82 | 62.99 |
| | PLNL | 94.44 | 83.62 | 58.10 | 82.10 | 67.62 | 72.11 | 76.33 |
| | CPL_RC | 90.67 | 85.36 | 51.89 | 71.54 | 58.33 | 64.71 | 70.42 |
| | CPL_CC | 93.39 | 88.12 | 51.95 | 73.61 | 60.53 | 67.46 | 72.51 |
| | CPL_LW | 90.83 | 85.72 | 52.19 | 71.70 | 58.45 | 64.79 | 70.61 |
| | **PMLL (Our)** | **95.94** | **92.01** | **64.89** | **89.48** | **78.92** | **80.91** | **83.70** |

## C.3 ADDITIONAL EXPERIMENTS ON PRIVACY-SENSITIVE CATEGORIES ACCURACY

Table 4 compares different methods on privacy sensitive categories under CLIP and Qwen generated PMLs. Our approach consistently achieves the highest accuracy across all datasets and both LVLM settings. Under CLIP generated PMLs PMLL improves average accuracy by more than twelve points over the best baseline demonstrating that our risk consistent estimator and privacy masked label strategy significantly reduce the noise in pseudo labels and prevent error accumulation. Under Qwen generated PMLs a similar trend is observed where PMLL achieves more than fourteen points gain over the second best method showing strong generalization across LVLM backbones. The gains are particularly large on fine grained datasets such as Food 101 and DTD indicating that our approach is especially effective when category boundaries are subtle and label noise is more harmful. These results confirm the robustness and cross model transferability of our framework.

## C.4 ADDITION COMPARISON ON TRAINING EPOCH

Figure 10 and Figure 11 spresent the training epochs for all methods across eight benchmark datasets. These results show that PMLL shows consistently faster convergence compared to all

Table 6: Test accuracy (%) of different size of privacy-sensitive categories. The best method is highlighted in **bold**.

| | | Caltech-101 | Oxford Pets | DTD | Flowers-102 | Stanford Cars | UCF-101 | Average |
|---|---|---|---|---|---|---|---|---|
| | DIRK | 92.17 | 81.39 | 47.81 | 69.23 | 59.72 | 64.68 | 69.17 |
| | PaPi | 91.60 | 82.69 | 47.22 | 68.78 | 60.08 | 66.85 | 69.54 |
| | SPMI | 85.72 | 74.35 | 42.55 | 59.07 | 36.66 | 52.82 | 58.53 |
| Size = 1 | PLNL | 93.02 | 82.07 | 50.12 | 70.85 | 59.60 | 65.32 | 70.16 |
| | CPL_RC | 90.67 | 85.34 | 51.89 | 71.54 | 58.34 | 64.71 | 70.42 |
| | CPL_CC | 93.38 | 88.28 | 51.95 | 73.61 | 60.48 | 67.46 | 72.53 |
| | CPL_LW | 91.24 | 85.31 | 52.19 | 71.70 | 58.50 | 64.79 | 70.62 |
| | **PMLL (Our)** | **95.34** | **90.65** | **60.11** | **83.03** | **71.17** | **75.28** | **79.26** |
| | DIRK | 91.85 | 81.68 | 49.29 | 68.33 | 60.02 | 64.18 | 69.23 |
| | PaPi | 92.09 | 82.83 | 48.40 | 69.18 | 60.75 | 65.61 | 69.81 |
| | SPMI | 85.56 | 73.78 | 43.97 | 59.12 | 36.65 | 53.42 | 58.75 |
| Size = 2 | PLNL | 92.94 | 81.77 | 51.12 | 71.34 | 60.07 | 64.58 | 70.30 |
| | CPL_RC | 89.95 | 85.31 | 51.59 | 70.95 | 57.56 | 64.13 | 69.92 |
| | CPL_CC | 93.34 | 87.88 | 51.44 | 72.79 | 59.92 | 66.48 | 71.98 |
| | CPL_LW | 89.95 | 85.01 | 51.44 | 71.22 | 58.34 | 64.77 | 70.12 |
| | **PMLL (Our)** | **95.25** | **89.70** | **60.23** | **81.00** | **71.29** | **75.55** | **78.84** |
| | DIRK | 91.93 | 81.11 | 49.65 | 68.17 | 59.64 | 65.37 | 69.31 |
| | PaPi | 91.72 | 82.56 | 47.75 | 67.07 | 60.81 | 66.27 | 69.36 |
| | SPMI | 85.72 | 73.32 | 43.91 | 57.82 | 36.80 | 53.11 | 58.45 |
| Size = 3 | PLNL | 93.10 | 81.71 | 51.60 | 70.04 | 59.77 | 65.13 | 70.23 |
| | CPL_RC | 88.98 | 84.55 | 51.07 | 70.80 | 57.28 | 63.83 | 69.42 |
| | CPL_CC | 93.09 | 86.96 | 50.81 | 72.12 | 59.86 | 66.18 | 71.50 |
| | CPL_LW | 89.01 | 84.78 | 50.84 | 70.79 | 57.40 | 64.70 | 69.59 |
| | **PMLL (Our)** | **95.30** | **90.27** | **59.34** | **82.01** | **70.86** | **74.81** | **78.77** |
| | DIRK | 92.41 | 81.47 | 49.53 | 70.20 | 60.44 | 65.64 | 69.95 |
| | PaPi | 92.21 | 82.80 | 49.00 | 69.59 | 61.22 | 66.77 | 70.27 |
| | SPMI | 86.09 | 73.26 | 42.79 | 60.29 | 36.79 | 53.64 | 58.81 |
| Size = 5 | PLNL | 93.35 | 82.64 | 52.01 | 73.04 | 60.59 | 65.48 | 71.19 |
| | CPL_RC | 88.31 | 83.88 | 50.49 | 70.05 | 56.89 | 63.10 | 68.79 |
| | CPL_CC | 92.55 | 86.19 | 50.03 | 71.43 | 59.40 | 65.56 | 70.86 |
| | CPL_LW | 88.45 | 84.01 | 50.27 | 70.05 | 56.73 | 63.89 | 68.90 |
| | **PMLL (Our)** | **94.32** | **91.17** | **58.27** | **83.11** | **71.84** | **75.95** | **79.11** |

baselines. Its training curve rises steeply in the early epochs, indicating that the model learns useful representations efficiently and reduces empirical risk quickly. While methods such as DIRK and PLNL exhibit slower and more gradual improvements, PMLL reaches a near-optimal region in fewer epochs and stabilizes earlier, demonstrating better optimization stability.

In terms of final accuracy, PMLL achieves the highest results across all datasets, suggesting that the proposed risk-consistent estimator not only accelerates learning but also improves generalization. The gap between PMLL and competing approaches becomes more pronounced in the later epochs, which highlights its ability to maintain low variance and avoid overfitting. These results indicate that our framework achieves a favorable tradeoff between efficiency and accuracy, making it well suited for practical privacy-sensitive learning scenarios where both convergence speed and final performance matter.

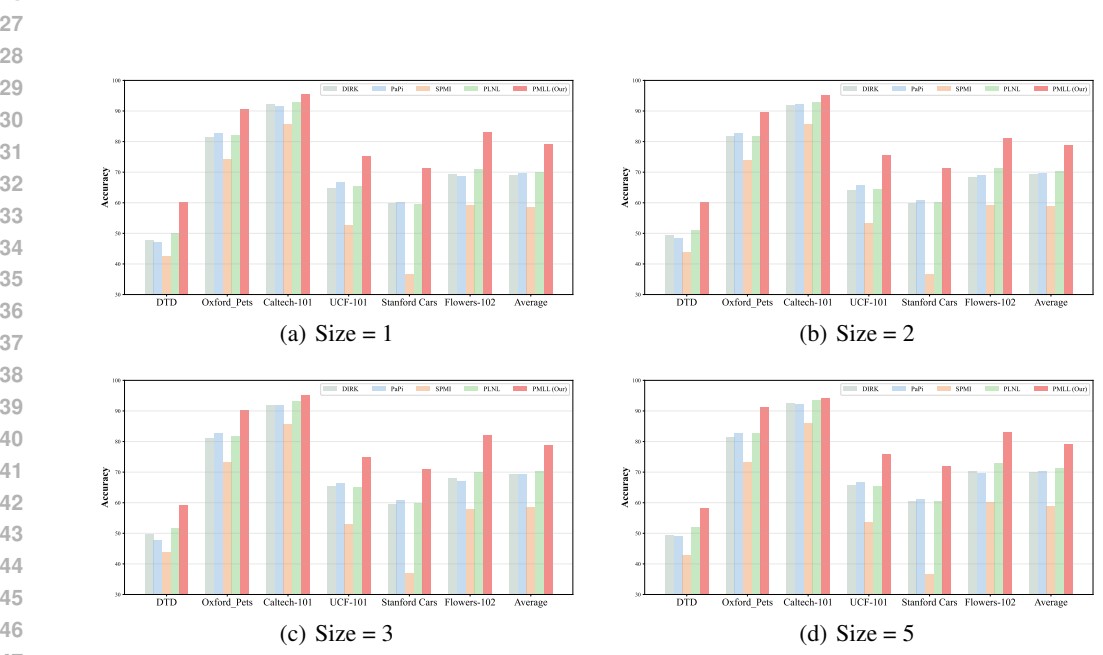

Figure 9: The classification accuracy of various methods with different size of privacy-sensitive categories using CLIP-generated PMLs.

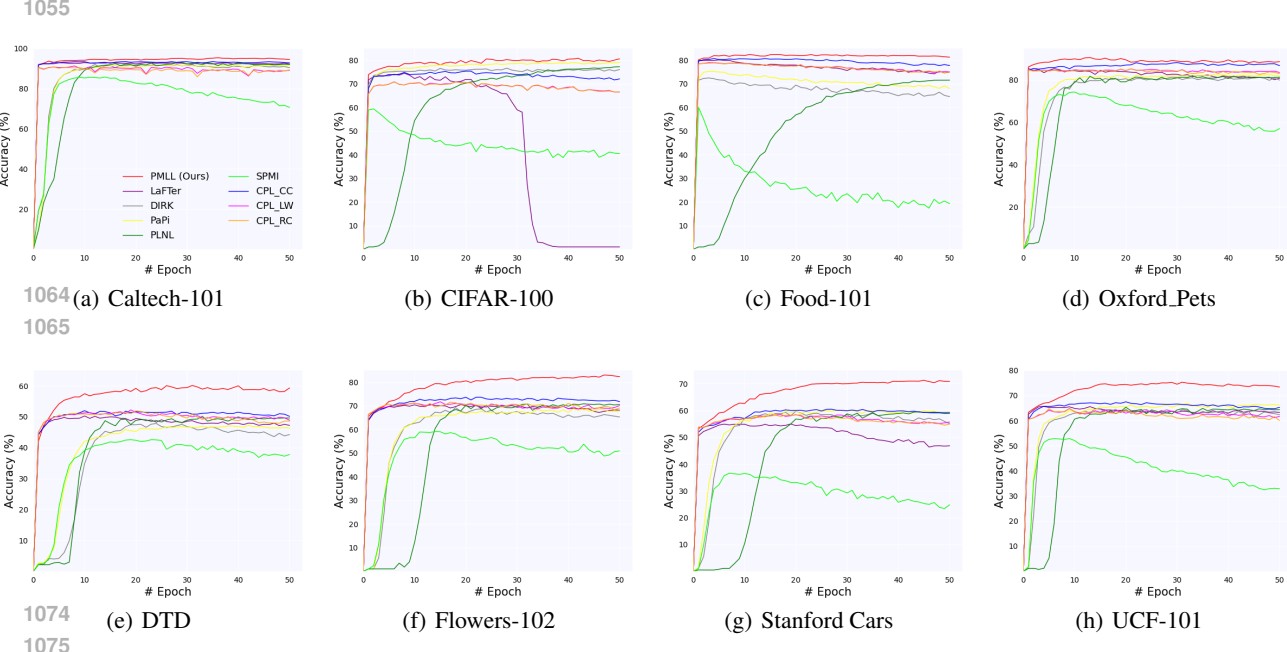

Figure 10: Training details of all epoch of various methods using CLIP-generated PMLs with ratio $q$ set to $0.1$.

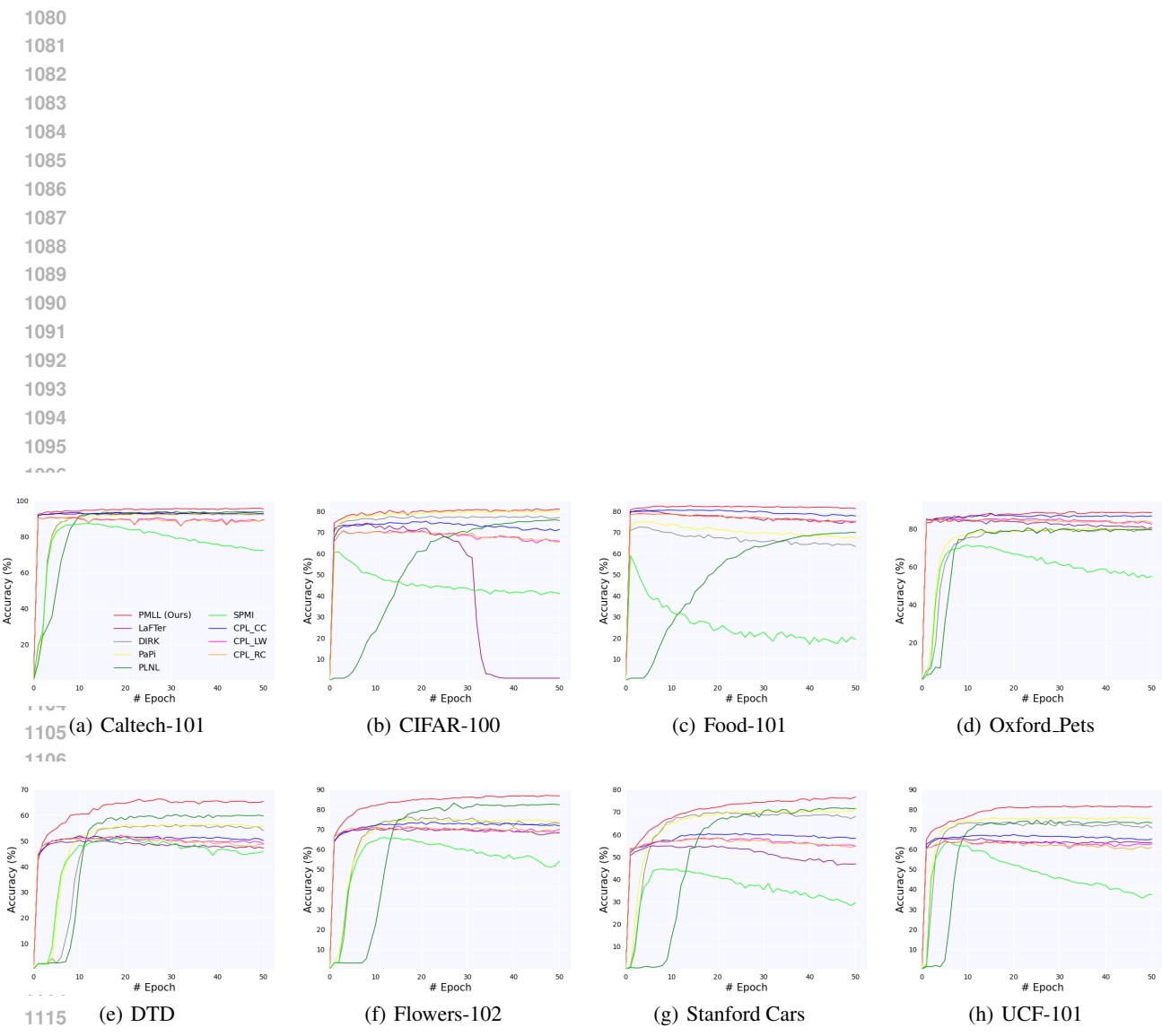

Figure 11: Training details of all epoch of various methods using Qwen-generated PMLs with ratio $q$ set to $0.1$.

