# OpenReview forum: "Preventing Privacy Leakage in Vision-Language Models: A Secure Framework for Large-Scale Image Classification"
_ICLR.cc/2026/Conference — ICLR 2026 Conference Withdrawn Submission_

### Official Review · Reviewer_Cpia · 2025-10-15

**Soundness:** 3
**Presentation:** 3
**Contribution:** 2
**Rating:** 4
**Confidence:** 4

**Summary:**

This paper addresses a practical privacy concern in using LVLMs for pseudo-label generation in image classification tasks.
It introduces a labeling framework PMLs that combines a fixed set of privacy-sensitive labels with a random subset of non-sensitive labels. Human annotators verify if the ground-truth is in this set. If not, LVLMs generate pseudo-labels only over the complement space, ensuring sensitive data is only exposed to human, not LVLMs.
Additionally, the authors propose a risk-consistent to train classifiers on this data, with conditional probabilities estimated via a convex combination of LVLM outputs and classifier softmax.
Experiments on eight benchmarks show superior performance over baselines like CPL and partial-label methods, approaching supervised accuracy while preserving privacy.

**Strengths:**

- The proposed method is simple and effective in the paper's setups
- Novel problem formulation
- Both empirical and theoretical are provided to demonstrate that the empirical risk approximates the true classification risk

**Weaknesses:**

- While the estimator is theoretically sound, the conditional probability estimation (convex combination with λ) is empirically motivated and lacks deeper justification. The paper notes it's "hard to estimate directly". This makes the method feel hybrid (theory for the risk, empirics for implementation) potentially limiting generalizability.
- While the privacy risks in LVLMs are timely, the proposed labeling framework and probability estimation (convex combination with λ) are straightforward and trivial.
- While the paper addresses the privacy risks, the experimental setups do not reflect the real-world privacy setting. Real-world sensitive datasets (e.g., medical) aren't tested, benchmarks like CIFAR-100 are clean but not privacy-laden.
- Only classification tasks are tested in this paper. Starting from image classification is good but the real-world problem in labelling data comes from other advanced tasks such as object detection, image captioning.

**Questions:**

- Have you considered extending PMLs to other tasks, like object detection or captioning?
- Why we only use the complement space when labeling with LVLM? and why not include $Y_{pl}$ since they are just text labels? Would providing all labels make LVLM predictions better?

---

### Official Review · Reviewer_ptib · 2025-10-30

**Soundness:** 2
**Presentation:** 1
**Contribution:** 2
**Rating:** 2
**Confidence:** 3

**Summary:**

This work introduces privacy-masked labels (PML) as a way to address emerging privacy issues in LVLM settings with an untrusted model host. In particular, assuming that the label of a given process is sensitive, PML learning (PMLL) restricts the untrusted model provider’s predictions to non-sensitive labels, producing pseudo-labels only for these (e.g., for KL-based learning of a classifier), thus limiting the provider’s exposure to the sensitive label. PMLs are thereby a combined mixture of private labels and random non-private labels. Importantly, if the ground-truth label of a data point is private, it is never sent to the untrusted model but instead directed to human annotators. The authors further present a risk-consistent estimator based on their setting as well as a performance-improving method. PMLs are then evaluated across eight classification datasets, consistently showing state-of-the-art performance (more on this below).

**Strengths:**

- LVLMs are heavily used in practice for label generation and will be used even more in the near future. Limiting the privacy impact of using these methods at scale is timely.
- In the respective setting PLML seems to provide strong utility numbers.
- The estimator derivation is  appreciated.

**Weaknesses:**

- The work in its current form has two main weaknesses in my opinion. The first is the presentation, which at times is unclear (smaller details mentioned below) and significantly affects the second point: the way human supervision is used in this technique. From a first read, it seemed that the definition of S (Eq. 2) creates a circular dependency on the ground-truth y to decide whether we need the ground-truth from a human (in which case, why bother) or whether we can obtain it via the LVLM (again, why bother, but at least this is not privacy-sensitive). This conflict is never clearly resolved in the paper. Even on the flipside, if this smaller binary decision were made by a human locally (e.g., to reduce their average workload), it would still require supervision of each individual sample. Also, for the case of “y not in Y,” we restrict the LVLM to essentially produce pseudo-labels across classes that we know to be wrong. Despite this, the method shows clear improvement over state of the art, which is at least odd and in practice should require an ablation of the individual effects of choosing this size.
- This also significantly affects the comparison with other methods in the evaluation, as they do not rely on ground-truth labels in a similar sense or involve partial human supervision for label generation (as a side note, the practical effort of this is not evaluated).
- The human effort at the current level seems to me roughly equivalent to simply labeling the dataset directly; otherwise, the framework would require a stricter analysis of the bias (and privacy risk) induced by potentially incorrect choices of “y in Y.”
- More fundamentally, the privacy angle (while featured in the title) seems to be not fully reflected throughout the work, even aside from not being part of the evaluation. In many cases, it is not typical in a privacy setting that the label alone defines whether data is private (e.g., a lung scan, the given example, might commonly be considered sensitive regardless of whether the resulting diagnosis is cancer or not). Such attribute-level issues are ignored here. Furthermore, the setting’s motivation, while justifiable to some extent, feels somewhat contrived, as the privacy issues discussed (especially in clinical data) are generally handled in the terms of service between the model provider and client. Most surprisingly to the reviewer, given the current state-of-the-art results compared to non-privacy-focused methods (notably those without the ground-truth caveat discussed above), the paper could stand more strongly on its performance than on its privacy angle (at least as the evaluation is currently structured).

## Smaller Weaknesses
- The figure captions are a bit short and do not provide much description of what is actually shown. Furthermore, Figure 1 is difficult to understand as the caption is unclear, the term “ordinary” is not previously used in the text, and it is unclear what a “label” refers to given that the image itself is shown.
- The overall structure of the work is somewhat hard to follow; the work first introduces a set of uncomputable elements and only for some of them explains how they are handled.

**Questions:**

In addition to the
- Based on the above, please specify exactly how your algorithm works, in particular the practical process of choosing y and determining S. Also, how do you restrict the LVLM to the set Y exactly?
- Does B2 assume that human labels are perfect?
- What would be the actual sensitive/private classes for any of your evaluation experiments?
- The practical case differentiates between S=0 (human label) and S=1 (machine), and this distinction is also ablated in Table 3. However, in the introduction of the evaluation section, it is written: “During training, the original labels of all images are replaced with Privacy-Masked Labels (PMLs).” What exactly is a PML in this case? In fact, Section 2.2 does not formally specify what a PML is: Is it (1) the restricted subset (referred to as the candidate set), (2) the VLM prediction on this set, or (3) the combination of either the VLM prediction or the human label?
- What are the practical sizes of the S=0 and S=1 subsets? How is this division handled for each of the compared methods? For example, if only one label is set as private, it seems that in many cases we would effectively be training on mostly pure ground-truth data.

---

### Official Review · Reviewer_522c · 2025-10-31

**Soundness:** 3
**Presentation:** 2
**Contribution:** 2
**Rating:** 4
**Confidence:** 3

**Summary:**

1. The paper identifies that conventional LVLM training and pseudo-labeling expose sensitive data categories, leading to potential memorization and leakage of private information.
2. The authors introduce an innovative framework that uses PMLs to prevent LVLMs from directly accessing sensitive labels. This is achieved by combining the sensitive set with a randomized non-sensitive subset.
3. The authors also develop a statistical estimator that ensures the model's learning process remains consistent and effective despite the restriction on using sensitive categories enforced by the PMLs.

**Strengths:**

1. The proposed PMLs framework fundamentally prevents LVLMs from accessing sensitive data and is compatible with various loss functions.
2. The proposed RCE is statistically proven to ensure valid and effective model training despite the enforced privacy constraints.
3. Experimental results demonstrate the method maintains competitive accuracy while guaranteeing robust privacy protection.

**Weaknesses:**

1. High Dependency on Human Labeling
The framework still mandates expensive human annotation for all high-risk (S=0) samples. Annotation costs could remain high, particularly if the proportion of sensitive data or required random labels is large.
2. Limitations in Defining and Enforcing Privacy
The privacy protection is fundamentally limited by a predefined, fixed sensitive label set. If this set is incomplete or fails to cover evolving definitions of sensitive data, the protection is compromised. Besides, the framework primarily prevents LVLMs from seeing the sensitive label. However, it does not account for the LVLM's powerful visual reasoning capabilities. If an image contains visually recognizable sensitive features (e.g., clear signs of a tumor, even if the label is hidden), the LVLM might still be able to infer the sensitive information from the image content itself, a risk not fully mitigated by the PMLs mechanism.
3. Fairness of Comparison
Methods like pseudo-labeling rely solely on model-generated labels. By contrast, the PMLs framework introduces an expensive human annotation process for all high-risk samples. It is intuitive that the inclusion of this high-quality human supervision would inherently enhance model performance, potentially making the comparison with pure pseudo-labeling baselines less rigorous.
4. Typos and Formatting Issues
The paper contains minor errors, such as a potential typo in Line 74 where "Figure 1(b)" might refer to "Figure 1(2)". The placement of figures (e.g., Figure 10 and Figure 11) extending beyond the page limit suggests a need for better organization and adherence to formatting standards.

**Questions:**

1. Could the authors please clarify the specific criteria and process used to define and select the sensitive label set across the different experiments? Specifically, given that the paper illustrates the privacy risk using pathological examples (suggesting a focus on clinical/personal data), but conducts experiments on common datasets like Food-101 and CIFAR-100. Are the same labels consistently designated as sensitive across all datasets, or does the definition change based on the dataset's inherent nature?For these non-pathological datasets (e.g., Food-101), what is the rationale for defining specific categories as privacy-sensitive within the scope of this work?
2. The core contribution lies on "Preventing Privacy Leakage." Could the authors provide direct empirical evidence (beyond classification accuracy) to validate the framework's effectiveness in preventing leakage?

**Details Of Ethics Concerns:**

The proposed framework explicitly relies on human annotators to label all samples designated as high-risk, which include sensitive categories. The paper is lacking sufficient details about this essential human annotation process, raising potential ethical concerns.

---

### Note · Authors · 2025-12-01

I have read and agree with the venue's withdrawal policy on behalf of myself and my co-authors.